# The Thousand Polish Genomes—A Database of Polish Variant Allele Frequencies

**DOI:** 10.3390/ijms23094532

**Published:** 2022-04-20

**Authors:** Elżbieta Kaja, Adrian Lejman, Dawid Sielski, Mateusz Sypniewski, Tomasz Gambin, Mateusz Dawidziuk, Tomasz Suchocki, Paweł Golik, Marzena Wojtaszewska, Magdalena Mroczek, Maria Stępień, Joanna Szyda, Karolina Lisiak-Teodorczyk, Filip Wolbach, Daria Kołodziejska, Katarzyna Ferdyn, Maciej Dąbrowski, Alicja Woźna, Marcin Żytkiewicz, Anna Bodora-Troińska, Waldemar Elikowski, Zbigniew J. Król, Artur Zaczyński, Agnieszka Pawlak, Robert Gil, Waldemar Wierzba, Paula Dobosz, Katarzyna Zawadzka, Paweł Zawadzki, Paweł Sztromwasser

**Affiliations:** 1MNM Bioscience Inc., Cambridge, MA 02142, USA; elzbieta.kaja@gmail.com (E.K.); 26adrian.l@gmail.com (A.L.); dawid.sielski@mnm.bio (D.S.); mateusz.sypniewski@mnm.bio (M.S.); wojtaszewska@gmail.com (M.W.); mmaria.stepien@gmail.com (M.S.); karolina.lisiak@mnm.bio (K.L.-T.); filip.wolbach@mnm.bio (F.W.); daria.kolodziejska96@gmail.com (D.K.); katarzyna.ferdyn@gmail.com (K.F.); maciej.dabrowski@mnm.bio (M.D.); alicja.wozna@mnm.bio (A.W.); paula.dobosz@gmail.com (P.D.); kasia@mnm.bio (K.Z.); pawel.zawadzki@mnm.bio (P.Z.); 2Central Clinical Hospital of Ministry of the Interior and Administration in Warsaw, 02-507 Warsaw, Poland; zbigniew.krol@cskmswia.pl (Z.J.K.); artur.zaczynski@cskmswia.pl (A.Z.); agnieszka.pawlak@cskmswia.pl (A.P.); robert.gil@cskmswia.pl (R.G.); waldemar.wierzba@cskmswia.pl (W.W.); 3Department of Medical Chemistry and Laboratory Medicine, Poznan University of Medical Sciences, 60-101 Poznan, Poland; 4Department of Genetics and Animal Breeding, Poznań University of Life Sciences, 60-637 Poznan, Poland; 5Institute of Computer Science, Warsaw University of Technology, 00-665 Warsaw, Poland; tgambin@gmail.com; 6Department of Medical Genetics, Institute of Mother and Child, 01-211 Warsaw, Poland; mateusz.dawidziuk@imid.med.pl; 7Biostatistics Group, Wrocław University of Environmental and Life Sciences, 51-631 Wrocław, Poland; tomasz.suchocki@gmail.com (T.S.); jszyda@gmail.com (J.S.); 8National Research Institute of Animal Production, 32-083 Balice, Poland; 9Institute of Genetics and Biotechnology, Faculty of Biology, University of Warsaw, 02-106 Warsaw, Poland; p.golik@uw.edu.pl; 10Department of Hematology, Frederic Chopin Provincial Specialist Hospital, 35-055 Rzeszów, Poland; 11Department of Neurology and Neurophysiology, Balgrist University Hospital, University of Zurich, 8008 Zurich, Switzerland; m.mroczek888@gmail.com; 12Department of Infectious Diseases, Medical University of Lublin, 20-059 Lublin, Poland; 13Department of Sports Medicine, Medical University of Lublin, 20-059 Lublin, Poland; 14Medical and Science Sp. z o.o., 08-455 Podebłocie, Poland; 15Institute of Human Genetics Polish Academy of Sciences, 60-479 Poznan, Poland; 16Faculty of Physics, Adam Mickiewicz University, 61-614 Poznan, Poland; 17Department of Internal Medicine, Józef Struś Multidisciplinary Municipal Hospital, 61-285 Poznan, Poland; marcin.zytkiewicz@gmail.com (M.Ż.); anna.bodora@gmail.com (A.B.-T.); welikowski@wp.pl (W.E.); 18Mossakowski Medical Research Centre, Polish Academy of Science, 02-106 Warsaw, Poland; 19Department of Hematology, Transplantation and Internal Medicine, University Clinical Center of the Medical University of Warsaw, 02-091 Warsaw, Poland

**Keywords:** variant, allele frequency, genome, whole-genome sequencing, population genomics, allelic distribution, Polish genomes

## Abstract

Although Slavic populations account for over 4.5% of world inhabitants, no centralised, open-source reference database of genetic variation of any Slavic population exists to date. Such data are crucial for clinical genetics, biomedical research, as well as archeological and historical studies. The Polish population, which is homogenous and sedentary in its nature but influenced by many migrations of the past, is unique and could serve as a genetic reference for the Slavic nations. In this study, we analysed whole genomes of 1222 Poles to identify and genotype a wide spectrum of genomic variation, such as small and structural variants, runs of homozygosity, mitochondrial haplogroups, and *de novo* variants. Common variant analyses showed that the Polish cohort is highly homogenous and shares ancestry with other European populations. In rare variant analyses, we identified 32 autosomal-recessive genes with significantly different frequencies of pathogenic alleles in the Polish population as compared to the non-Finish Europeans, including *C2*, *TGM5*, *NUP93*, *C19orf12*, and *PROP1*. The allele frequencies for small and structural variants, calculated for 1076 unrelated individuals, are released publicly as The Thousand Polish Genomes database, and will contribute to the worldwide genomic resources available to researchers and clinicians.

## 1. Introduction

An individual genome carries over four million single-nucleotide variants, small insertions and deletions, and thousands of structural variants. These alterations in DNA are responsible for 0.1% of the difference between the genomes of two unrelated individuals [1] and lead to phenotypic variation among people, affecting disease susceptibility, drug response, and physical traits such as height [2]. Mapping naturally occurring (1) [1] and pathogenic [3] genetic variation is, thus, critical for studying human health and disease.

Since the completion of the Human Genome Project, numerous efforts have been made to analyse human DNA, creating open, population-scale databases of human variation [1,4,5,6,7]. These reference datasets are invaluable for filtering common variation in rare disease and GWAS studies; for imputation of variants; and for studying human diversity, evolution, and migration, having laid a foundation for precision medicine [8,9,10,11].

Despite the sheer size of these reference databases, the information on the full spectrum of the world’s genetic variation remains incomplete [12]. The gaps are successively being filled by projects focusing on human genome diversity [13,14], building local variome databases [15,16,17,18], or even creating population-specific reference genomes [15,19,20]. Besides fostering basic research, these efforts establish local genomic resources that enable customisation of medical genetics services to the needs of particular populations, for instance by prioritising and tailoring genetic screening programmes or improving local guidelines for genetic counselling.

The Slavic populations account for over 4.5% of the world’s inhabitants, yet no centralised, open-reference database of genetic variation of any Slavic population exists to date. Representation of genomes from Central and Eastern Europe are also scarce in the world’s reference databases. In response to this need, we used whole-genome sequencing data from 1222 Polish individuals, including over 120 families, to create such a resource. High-quality and depth of the sequencing data enabled us to build a unique repository of genetic variation in the Polish population—The Thousand Polish Genomes database (see Web Resources). We characterised a wide spectrum of genetic alterations, including (among others) small and structural variants, and compared these to other variant datasets from continental and European populations, focusing primarily on clinically relevant differences. We hope that the controlled-access allele frequency database released as a result of this study will become a valuable resource for clinical and population genetics, biomedical research, and demographic inference in Europe and worldwide.

## 2. Results

### 2.1. Characteristics of the Cohort

The whole cohort consisted of 1222 individuals of Polish origin, 1076 of whom were unrelated. The median age of participants was 45.4 (2–99) years, with predominance of males (697 vs. 525). Figure 1B shows the age distribution of the unrelated participants. The analysis of clinical data showed that the most common chronic diseases reported by the participants were hypertension (13%), cancer (4.6%), diabetes (4%), and hypothyroidism or Hashimoto’s disease (3%). No health problems (excluding COVID-19 infection) were reported by 86% of participants.

### 2.2. Genetic Variation in the Polish Population

We processed sequencing data from 1222 individuals, totalling over 1018 billion read-pairs, and yielding an average 35.26× read depth per genome (Appendix A). In every sample, over 91% of the reference genome was covered with at least 10 unique mapping reads. We detected and genotyped small and structural variants, runs of homozygosity (Appendix A), mitochondrial haplogroups (Appendix A), and Mendelian inconsistencies (Appendix A). Except for Mendelian inconsistencies, in all the analyses we used genomes of the 1076 unrelated participants (POL cohort hereafter). The allelic frequencies of small and structural variants identified in this cohort were released openly for academic and clinical research as The Thousand Polish Genomes database (see Web Resources).

#### 2.2.1. Small and Structural Variants

In 1076 unrelated individuals, we identified 33.4 M single-nucleotide substitutions and 5.9 M small insertions and deletions. An individual genome contained on average 3.71 M (3.59–3.81) single-nucleotide substitutions, of which 2.22 M (1.97–2.37) were heterozygous and 1.49 M (1.38–1.63) were homozygous. Together with on average 0.76 M (0.74–0.78) short insertions and deletions, we found 4.48 M small variants per individual (Appendix A), of which 15,877 were private variants (singletons).

Structural variants, typically defined as rearrangements of >50 bp of the DNA sequence, were also analysed. The Thousand Polish Genomes database holds 76,584 structural variant calls, of which 23,076 were marked as high quality (19,808 deletions, 2270 duplications, and 998 inversions; Appendix A). The majority of the deletions were shorter than 1 kb (63.6%; 93.1% were <10 kb), with peaks at 50 bp and 300 bp (Appendix A). Distribution of duplication lengths also showed two peaks, at 200 bp and near 10 kb. Only 10 duplications (0.5%) exceeded 1 Mb in length. Inversions were the rarest category of the three structural variant types detected. The great majority were below 10 kb (peak at 1–2 kb; Appendix A), but they displayed the highest fraction of events longer than 1 Mb (2.7%; *n* = 10) among the three SV types. An average individual genome carried 2598 large deletions (2430–2721), 106 duplications (87–129), and 76 inversions (57–655) (Appendix A). The median length of the genomic sequence affected by SVs in a single individual was 12.6 Mb (2.94 Mb, 2.17 Mb, and 7.44 Mb for deletions, duplications, and inversions, respectively), exceeding the total DNA length affected by small variants in any individual.

#### 2.2.2. Distribution of Allele-Frequencies

We compared the distributions of allele frequencies among different variant types, including substitutions, short indels, and high-quality structural variants. The results presented in Figure 2 and Table 1 indicate that the majority of substitutions (SNVs) (52%) were private variants, observed in a single individual (MAF ~0.05%). For large deletions and duplications, singletons constituted 41.9% and 44.8%, respectively, while the inversions constituted 67.1%. Among all analysed variant categories, singletons were the least abundant in indels, constituting 34.5% of all short insertions and deletions.

Common (MAF ≥ 1%) variants in our cohort were most abundant among indels (37.9%), deletions (32.7%), and substitutions (28.2%), and clearly less abundant among duplications (14.8%) and inversions (19.2%). This trend was observed throughout the entire common variant frequency spectrum (MAF 1–100%). Fixed alleles (allelic frequency equal to 100%) were present among all types of variants with similar rates, constituting 0.1–0.5% SVs, 0.5% substitutions, and 0.36% indels.

### 2.3. Relationship of the Polish and Global Populations

We explored the diversity of the Polish population and relationships between POL, continental populations, and European sub-populations from the 1000 Genomes project. Using common variants (MAF > 1%), we performed a principal component analysis (PCA), calculated Fst statistics, and performed density-based spatial clustering (DBSCAN). In a comparison with continental populations, we observed that the POL cohort was homogenous and clustered within the European population (Figure 3A,B and Appendix A; average Fst 0.002). Considering the Fst statistic, POL shared the least similarity with the African and East Asian populations (average Fst values of 0.024 and 0.021, respectively). An average Fst statistic calculated over sliding windows of 1000 SNPs is presented in Appendix A. In a PCA of the European subpopulations, virtually all POL samples (1065 out of 1076) were clustered with other European ancestries: 1063 individuals with the CEU, 2 with the TSI subpopulation, and 11 with non-European populations (Figure 3C,D). The average Fst statistics for European subpopulations ranged from 0.002 (CEU) to 0.009 (TSI and FIN; Appendix A). The results of these three analyses agreed in demonstrating that the POL cohort is highly homogeneous and that participating individuals belong to the European populations, both in terms of continental and sub-populational clustering.

We also performed admixture [21] analysis in the unsupervised mode with the 1076 unrelated individuals from POL and the 1000 Genomes dataset, either with only five European populations (Appendix A) or with the entire global dataset (Appendix A). For the European dataset, the number of ancestral populations (K) was estimated to be 2 based on the minimal cross-validation error or 3 based on the MedMeaK, MaxMeaK, MedMedK, and MaxMedK estimators (Puechmaille, 2016), which overcome problems related to uneven sampling, whereas for the global dataset K = 8 was chosen based on the same criteria. The Polish cohort was mostly homogeneous and dominated by one ancestral component, which was also present in the populations from Western (GBR and CEU),and (for K = 2 in the European dataset) Northern (FIN) Europe, but absent from the Southern (IBS and TSI) European populations, which were remarkably uniform. The ancestral component found in IBS and TSI was also present in the POL cohort, albeit at a lower proportion, with the exception of a few individuals.

### 2.4. Functional Impact of the Variants

#### 2.4.1. Impacts of Small Variants on Protein Function

Depending on the impact on the genomic sequence, variants were distributed non-uniformly across the allele-frequency spectrum. Variants in protein-coding regions were depleted in higher frequencies (Appendix A) compared to non-coding variation (e.g., intergenic, intronic). Larger differences were observed among exonic variants (Figure 4), where protein-altering variation (nonsense variants, frameshift indels, and missense variants) was associated with lower allele frequencies in contrast to UTR, synonymous, and non-exonic variants. A summary of variant counts, SNVs, and indels, divided into three population frequency tiers and categorised by functional impact, is presented in Table 1.

#### 2.4.2. Disease Causing Variation

In the POL cohort, we identified 808 rare variants (MAF < 0.1% in gnomAD v3.1.1) reported in ClinVar (see Web Resources) as pathogenic or likely pathogenic. The selected variants were divided into four Clinvar confidence tiers (see Methods): (1) 309 variants with a low confidence level; (2) 473 variants with a moderate confidence level, for example rs11555217 in *DHCR7* gene, described in more details in Appendix A; (3) 23 pathogenic variants with high confidence; (4) 3 variants with the highest confidence. In total, 591 (73.1%) of these pathogenic or likely pathogenic variants were private and 217 variants were carried by 2–26 individuals, e.g., variants in *PAH*, *OTOG*, and *CBS*. Selected mutations and genes are presented in Appendix A.

Among the pathogenic variants, only a small set is medically actionable. We sought to estimate the rate of actionable variation in the POL cohort following the latest (v.3.0) American College of Medical Genetics and Genomics (ACMG) recommendations [22]. Using the list of 73 medically actionable genes selected by ACMG, we identified 50 variants in POL with a high impact on the encoded protein. After manual curation following the ACMG guidelines, 15 variants were assessed as pathogenic and 3 as likely pathogenic, affecting 20 of 1076 (1.9%) unrelated individuals in our cohort. Only two out of the 20 participants carrying the pathogenic and likely pathogenic variants showed reported disease symptoms related to the mutated gene. In total, 17 (94%) of the variants were private to a single person and were mostly related to cancer predisposition genes such as *MUTYH*, *MSH6*, *BRCA2*, and *PALB2*. In one case, a variant was carried by three individuals (NM_007294.4(BRCA1):c.5266dup (p.Gln1756fs); rs80357906; MAF = 0.14%), showing much higher allelic frequency than in the gnomAD NFE population (0.02% in gnomAD 2.1.1; 0.006% in gnomAD 3.1.1). Despite this, we did not observe significant enrichment of pathogenic *BRCA1* variants in POL vs. NFE (0.23% vs. 0.06%; Fisher *p*-value 0.06).

#### 2.4.3. Pathogenic Allele Burden

To assess the pathogenic allele burden in the Polish population, we performed a similar analysis as for *BRCA1* above, systematically for 2463 genes associated with autosomal recessive (AR) diseases from the OMIM (see Web Resources) database. Small variants located within selected AR genes and in +/− 5 kb regions from the start and end of their coding DNA sequence were selected. Of those, 841 (POL) and 11,255 (gnomAD) variants were annotated in the ClinVar database as pathogenic or likely pathogenic. Per-gene analysis of the cumulative allele frequencies of pathogenic alleles showed significant differences (Fisher q-score < 0.05) in 32 genes between POL and gnomAD NFE populations (Appendix A, Figure 5). In 23 genes, the cumulative frequency of pathogenic variation was higher in the POL cohort than in NFE, and in 9 genes it was lower. In 7 out of these 9 genes, no pathogenic alleles were found in POL, with a marked example of *ACKR1* carrying a single pathogenic variant with 1.6% MAF in gnomAD and none in POL. Among the genes with a cumulative frequency significantly lower in POL than in NFE, only two carried any pathogenic alleles in POL, i.e., *HTT* (1 variant in POL; MAF 0.2% in POL and 1.3% in NFE; q-score 4.7 × 10^−5^) and *PADI3* (2 variants in POL; cumulative frequency 0.8% in POL and 1.8% in NFE; q-score 0.01). The variant in *HTT* (RCV001335909.1) was removed from Clinvar after the release we used for the analysis.

Among the top differing genes, we noted *C2*, a gene associated with primary immunodeficiencies (C2 deficiency; OMIM #217000). It ranked as the most enriched in pathogenic alleles in POL, with a cumulative frequency 1.7-fold higher in POL than in NFE, and a single contributing variant NM_000063.6(*C2*):c.841_849 + 19del (rs9332736; MAF 2.5% in POL; 0.72% in NFE 2.1.1; 0.75% in NFE 3.1.1). Although in the majority of populations available in the dbSNP database the variant had an MAF below 1%, in Estonia and Northern Sweden it reached 1.8% and 1.3%, respectively, which may suggest higher frequency levels in Central and Northern Europe. Among other top differing genes, we identified founder mutations originating from Poland or Central and Eastern Europe in *TGM5* (c.337G > T), *NUP93* (c.1772G > T), *C19orf12* (c.204_214del), and *PROP1* (c.301_302del and c.150del) [23,24,25,26].

We attempted a complementary analysis using 1668 and 2622 rare (AF < 1%), putative LoF structural variants in the AR genes in the POL and gnomAD-SV EUR cohorts, respectively. We first looked at the overlap with the genes identified as significant in the pathogenic allele burden analysis above. Only 3 of them had a difference in SV frequencies exceeding 0.5%, i.e., *SBF2* (0.4% LoF SVs in POL and 1.3% in gnomAD), *NUP93* (0.93 and 0.08%), and *COL18A1* (0.98 and 0.16%). Overall, for 3 genes we found absolute differences in cumulative MAF exceeding 2% (Appendix A), with an outstanding case of *MTMR2* having a cumulative frequency of LoF SVs above 14.8% in POL. A detailed examination of the genes revealed an accumulation of 13–24 overlapping heterozygous SVs in a total of 24 individuals. If these individual SVs were genotyped as one complex SV, the frequency of the variant haplotype would have been ~1%. In the gnomAD-SV EUR, the cumulative frequency of LoF SVs in *MTMR2* was 0.05%, with no complex SVs present in this gene (described in more details in Appendix A).

Next, we analysed 248 gene panels from the Genomics England PanelApp website (see Web Resources), containing a total of 1823 AR disease genes. We identified significant differences (q-score < 0.05) in cumulative allele frequencies between the POL and NFE populations in 51 gene panels (Appendix A), 8 of which were enriched and 43 depleted in pathogenic alleles in POL. The panels significantly enriched in POL were smaller (2–44 genes; median = 11, mean = 14) compared to panels enriched in NFE (1–2415; median = 40, mean = 235). The top three panels with enrichment of pathogenic alleles in POL (“peeling skin syndrome”, “Epidermolysis bullosa”, and “Epidermolysis bullosa and congenital skin fragility”) were related to skin disorders, and all contained the *TGM5* gene described above. Also in “optic neuropathy”, the fourth most enriched panel in POL, the majority of pathogenic alleles came from one gene—*C19orf12*, with NM_031448.6(C19orf12):c.171_181del (rs515726204) being a founder mutation originating from the Polish population [25]. Additionally, we did not find any specific relation between the top differing genes and disease panels. The panels with greatest depletion of pathogenic alleles in POL were “hypotonic infant”, “neuromuscular disorders”, and “paediatric disorders”.

## 3. Discussion

Large-scale whole-genome sequencing (WGS) projects of human samples have proliferated during the last decade, with flagship initiatives such as the “100,000 Genomes” project in the UK; “1+ Million Genomes” initiative in Europe; and “GenomeAsia 100 K Project”, “3MAG Project”, and “All of Us” in the United States continuously raising the participant numbers. Population-scale WGS of local communities (e.g., country or region) has also been performed, including tens of hundreds of participants from the Netherlands [16], Iceland [27], Sardinia [18], and Turkey [15]. Such initiatives emphasise the importance of screening variation in diverse populations, most-often missing from the largest databases [15], as well as building local genomic resources for clinical genetics and fostering research that was not possible before.

In this study we present the results of whole-genome sequencing and comprehensive analyses of a wide spectrum of genomic variation in 1222 individuals of Polish ancestry. We detected and genotyped small and structural variants (SV), runs of homozygosity, mitochondrial haplogroups, and *de novo* variants. We also analysed the population structure and identified disease-causing alleles enriched and depleted in the Polish population. Allele frequencies of the unrelated individuals were released as a publicly available resource, The Thousand Polish Genomes database. This is the first effort of this scale for a single-country population in Central Europe, complementing the international efforts to map human genomic variation. We hope the database will equip researchers and medical geneticists with a so-far-missing freely available reference dataset of Slavic variant frequencies.

A cohort of this size enabled analysis of genetic similarities of Poles and other populations. Considering both mtDNA haplogroups (Appendix A) and common variants, the POL cohort was found to be highly homogenous, and as anticipated sharing ancestry with the other European populations. The admixture analysis indicated that the POL population is dominated by a single ancestral component that can also be found in GBR, CEU, and possibly FIN populations, albeit at a lower percentage (Appendix A). Due to the lack of other samples from Central and Eastern Europe in the 1000 Genomes dataset, making any detailed inference about the ancestry of the Polish population compared to the rest of the continent is currently impossible. These results are, however, consistent with the idea that the genetic composition of the populations from Central and Eastern Europe was shaped mostly by the steppe pastoralist migrations in the Bronze Age [28]. The steppe ancestry formed a gradient of decreasing admixture into Northwestern Europe and had less influence on the Southern Mediterranean populations. Our data are consistent with this scenario, but the populations available in the 1000 Genomes dataset are not sufficient to test this hypothesis. A better sampling of WGS genomic data from Central and Eastern Europe, as well as from the Caucasus and Western Asia, would be required to reconstruct the history of this part of the world.

The sequencing data give a unique opportunity to interrogate rare variation. The rare alleles are considered relatively new in the population [29] and they tend to cluster geographically more than the common alleles [1,30,31], possibly giving rise to differences in inherited disease prevalence. Characteristics of runs of homozygosity in POL did not suggest increased risk of recessive diseases caused by homozygosity when compared to European population data [32,33]. However, we noted considerable differences in the frequencies of individual variants, and cumulatively in gene burden and gene panel analyses, when compared with non-Finish European populations. Variants in some of these genes were associated with the founder’s effect originating in Central Europe (e.g., *TGM5* [24], *PROP1* [26], *C19orf1* [25], *DHCR7* [34], *NBN* [35]), or reported before as frequent in the Polish population, e.g., rs80357906 in *BRCA1* [36], being a founder mutation in Ashkenazi Jews [37,38]. Others, despite showing considerable differences in frequency, have not been described before (e.g., *ACKR1*, *C2*, or *MASP1*). Further research is needed to investigate if this was due to a lack of comprehensible screening of these genes and phenotypes in the Polish population [24], incomplete penetrance of the variants, or incorrect pathogenicity status in Clinvar. The latter cannot be excluded, as two variants in the top differing genes were removed from Clinvar since the release we used for variant annotation.

Ideally, the analysis of pathogenic allele burden should be performed including frequencies of loss-of-function structural variants (SV). SVs collectively affect a larger part of the genome than the small variants [39], which we also observed in our results, and they may contribute disease-causing alleles, in some cases with greater frequency than small variants [40]. Due to the lack of common reference data (gnomAD NFE vs. gnomAD-SV EUR), we performed the SV burden analysis separately, identifying genes with significantly different cumulative frequencies of LoF-SV alleles. Our results pinpoint genes and phenotypes where an SV component should be considered when designing screening assays, as interrogating only small variants may be insufficient in capturing the full spectrum of the cases.

However, we must acknowledge the effect of technical differences in SV detection. While for small variants the differences in top-performing variant callers are modest and allelic frequencies between datasets are comparable, pipelines for detection of structural variants are known to provide less coherent results [41]. In our study, this was illustrated well for *MTMR2* complex variants (Appendix A). In a perfect study scenario, the analysis we attempted would require strict harmonisation of SV detection methods between the datasets, and preferably also library preparation protocols, as SV detection can be sensitive to the insert length [41]. Compared to other large-scale studies of SVs [42,43,44], we report lower SV counts per individual and larger fractions of affected genomic sequences. The differences may stem from larger divergence of the Central European populations from the reference genome compared to Western Europeans, but the above-mentioned methodological differences have also contributed. A single-tool approach with joint genotyping of the SVs, which we used, could have resulted in higher false-positive rates and lower sensitivity than ensemble methods used in other studies [39,42]. Considering the uniqueness of this data and the role it could play, for instance in rare variant filtering when the same SV detection methods are applied, we decided to release the unfiltered SV frequencies. We also note that further analyses combined with populational aCGH (array-comparative genomic hybridisation), long-read sequencing, or optical mapping are needed to explore and validate the diversity and clinical utility of SV patterns in the Polish population, as their pathogenic potential and involvement in complex genetic traits are still poorly understood.

It must be noted that the POL cohort is not representative of the entire Polish population. The participants were recruited to study the genetic susceptibility to COVID-19 infections; therefore, we did not focus on collecting samples from Polish minorities such as Kaschubians, Karaims, Lemkos, or Polish Jews. We also did not correct the much higher proportion of donors recruited from municipal areas compared to rural areas, nor the proportions of individuals from different voivodeships that are skewed in favour of Masovia and Greater Poland. As a result, the far eastern voivodeships were underrepresented, which could have led to missing variants native to the Tatar or the Russian populations. However, this is the largest whole-genome screening of the Slavic and Central Europe populations done to date, and one of the largest controlled-access allele frequency datasets generated with high-coverage WGS data, which have already proven a very useful resource for research and clinical genetics in the region.

The genomic analyses presented in this manuscript are far from exploring the full depth of the covered topics and should be taken as an invitation for further research. Our main goal was to generate a dataset and release an allele frequency database to researchers around the world, filling in a gap in the genomic map of Europe. Nevertheless, our results lay the foundation for further research in the population history and epidemiology of diseases caused by mutations in the autosomal-recessive genes and offer opportunities for tailoring NGS-based genetic screening tests and guidelines for clinical geneticists in Poland. We also suggest that national guidelines available in the Polish language should be issued to address the importance of genetic counselling for clinical WES and WGS results concerning incidental findings specific to our population, as availability to whole-genome analyses will successively increase.

## 4. Materials and Methods

### 4.1. Donors’ Characteristics and Sample Collection

The studied population consisted of participants recruited during the “Search for Genomic Markers Predicting the Severity of the Response to COVID-19” project related to genetic predisposition to COVID-19 severity. Samples were collected from 1222 individuals of Polish origin between April 2020 and April 2021. The whole cohort comprised three groups: (1) The first group comprised 546 participants recruited by the Central Clinical Hospital of the Ministry of Interior and Administration in Warsaw. The majority of these individuals (496 out of 546) were patients suffering from COVID-19. Peripheral blood samples from this group were collected during hospitalisation. (2) The second group comprised 646 individuals who were self-enrolled volunteers for the project. Samples from this group were collected in blood collection facilities from all over Poland, including almost all local Polish administrative units (15 out of 16 voivodeships; Figure 1A). (3) The third group comprised 30 individuals who were recruited in the Jozef Strus Multidisciplinary Municipal Hospital in Poznań from the cohort suffering from COVID-19. The study participants were unrelated individuals (*n* = 1076) or families (*n* = 126). Among participating families, there were 73 trios, 12 quartets, 7 pairs of siblings, and 34 families consisting of one parent and a child. The genomes of 1076 unrelated participants were used for most of the analyses presented in this study and are referred to as the POL cohort herein.

Basic clinical data were collected from all participants, including data on gender, age, BMI, and comorbidities (diabetes, hypertension, ischemic heart disease, stroke, heart failure, cancer, kidney failure or disease, hepatitis B, chronic obstructive pulmonary disease). For some participants, additional clinical data (genetic disorders, flu vaccination status, tuberculosis and measles vaccination status, smoking habits, hepatitis C infection) were collected and are available upon request. Only individuals without diagnosed severe genetic disorders were qualified for the study.

### 4.2. Compliance with Ethical Standards

All participants, or guardians or parents for the participants under 18, provided informed consent before collection of blood samples and filling in the clinical data form, which included a questionnaire about country of origin and chronic diseases. The ethical approval for the study was obtained from the Ethics Committee of the Central Clinical Hospital of the Ministry of Interior and Administration in Warsaw (decision nr: 41/2020 from 3 April 2020 and 125/2020 from 1 July 2020). The study complied with the 1964 Helsinki Declaration and its later amendments and adhered to the highest data security standards of ISO 27001 and the General Data Protection Regulation (GDPR) act.

### 4.3. Total Quality Management Utilised in the Study

The project was carried out in accordance with the total quality management (TQM) methodology, which ensures the quality of results. The TQM requires the definition of all critical points of the procedures, such as reference ranges for collected biological materials, preparation and isolation procedures, DNA concentration and quality information, and genomic sequencing procedures, including quality control of the data. The legal and ethical transparency of the entire project was ensured, including the confidentiality, integrity, and impartiality of the data. Additionally, a risk analysis was performed, possible difficulties were mapped, and corrective actions were planned.

### 4.4. Sample Collection and Whole-Genome Sequencing

Here, 4 mL samples of K-EDTA peripheral blood from participants were collected according to a standardised quality management system protocol. Genomic DNA samples were isolated from the peripheral blood leukocytes using a QIAamp DNA Blood Mini Kit, Blood/Cell DNA Mini Kit (Syngen, Bengaluru, India), and Xpure Blood Kit (A&A Biotechnology, Gdańsk, Poland) according to the manufacturer’s protocols. The concentration and purity of isolated DNA were measured using the NanoDrop^TM^ spectrophotometer, while the DNA integrity was evaluated via electrophoresis. The sequencing library was prepared by Macrogen Europe (Amsterdam, the Netherlands) using a TruSeq DNA PCR-free kit (Illumina Inc., San Diego, CA, USA) and 550 bp inserts. The quality of DNA libraries was measured using a 2100 Bioanalyser from Agilent Technologies. Subsequently, whole-genome sequencing (WGS) was performed on the Illumina NovaSeq 6000 platform using 150 bp paired-end reads, yielding an average read depth of 35.26× in the cohort (Appendix A).

### 4.5. WGS Data Processing

The quality of sequenced reads was assessed using FastQC v0.11.7 (see Web Resources) and reads were subsequently mapped to the GRCh38 human reference genome using the Speedseq framework v.0.1.2 [45], encompassing alignment with BWA MEM 0.7.10 [46], SAMBLASTER v0.1.22 [47] duplicate removal, and Sambamba v0.5.9 [48] sorting and indexing. The read depth was calculated using Mosdepth v0.2.4 [48], considering only reads with MQ > 0. Single-nucleotide variants and short indels in the nuclear genome were detected using DeepVariant v0.8.0 [49] and jointly genotyped with GLnexus v1.2.6-0-g4d057dc [50]. Next, multiallelic variant calls were decomposed into monoallelic and normalised groups using BCFtools v1.9 [51]. Structural variants were called and jointly genotyped using Smoove v.0.2.6 (see Web Resources).

Variants were annotated using Ensembl Variant Effect Predictor v.97 [52], including references to databases of genomic variants from ClinVar v. 201904 [3] and dbSNP build 151, variant population frequencies from the 1000 Genomes Project [53], and GnomAD v2.0.1 and v3.0 [6], as well as pathogenicity scores, including Polyphen-2 [54], SIFT [55], and DANN [56]. All WGS data processing was automated with a Ruffus framework [57], and selected analysis steps were parallelised with the GNU parallel tool [58].

### 4.6. Variant Analysis

Small variants of 1076 unrelated individuals were extracted from the normalised multisample VCF file using BCFtools [51]. Sites with genotyping rates below 90% and invariant in the group were excluded. The resulting dataset, with an average call rate per individual of 99%, was used in all subsequent analyses. Jointly called structural variants were also filtered for related individuals, and subsequently annotated and filtered according to the Smoove author recommendations (see Smoove in Web Resources), i.e., heterozygous calls with MSHQ < 3, deletions with DHFFC ≥ 0.7, and duplication calls with DHFFC ≤ 1.25 were excluded from the analysis. We also required at least one split-read per allele in the cohort. Structural variants passing these filtering criteria are referred to as high-quality in the manuscript. Break-end events (BND) listed in the Appendix A were omitted from the main results section due to their uncertain biological significance. Variant statistics were generated with BCFtools [51] and custom-written bash scripts. Pathogenic variant classification was performed using the Clinvar “four star” scheme [59]. Data processing and visualisation was programmed in R [60] using the tidyverse package [61]. VCF files with small and structural variants along with allele frequencies were released as publicly available resources—The Thousand Polish Genomes database (see Web Resources).

### 4.7. Population Analysis

We evaluated the homogeneity of the Polish population (POL) using three steps. In the first step, we checked whether all analysed POL samples belonged to the European population. We used variants common between the Thousand Genomes Project (1000 G) and our cohort data. To minimise technical bias, we did not use original 1000 G variant files, but instead variants provided by Yun et al. (2021) [50], obtained using the same version of the variant caller as in our project. The variants were pruned to MAF over 1% and to be in linkage equilibrium. In the second step, we performed a principal components analysis (PCA) on the 1000 G dataset and projected the data into POL genotype data. We trained a random forest model with the first six principal components (PCs) on the 1000 G dataset and predicted the ancestry of POL individuals. In the third step, in order to find out which European population is the closest to POL, we used the subpopulations information available in the 1000 G data set (Utah Residents (CEPH) with Northern and Western European ancestry (CEU), Finnish in Finland (FIN), British in England and Scotland (GBR), Iberian in Spain (IBS), Toscani in Italy (TSI)) and repeated the PCA and prediction based on random forest analyses as described above.

We also performed population structure and ancestry analyses using ADMIXTURE [21] with the unrelated individuals from our cohort, filtered and pruned as described above, either with the entire 1000 G dataset or with the five European populations. The analysis was performed in the unsupervised mode for a number of ancestral populations (K) ranging from 2 to 10 or 20, for the European or world populations, respectively. Cross-validation error, as well as the MedMeaK, MaxMeaK, MedMedK, and MaxMedK estimators [62], were used to determine the best value for K using the StructureSelector interface [63].

To estimate the average fixation index between two groups of individuals from SNPs, we used Fst statistics calculated using the formula proposed by Weir and Cockerham [64]. The calculations were performed with PLINK v1.90b6.21 software [44]. We also compared the Polish population with all populations available within the 1000 G database and checked whether Fst patterns were consistent with the continental and populational clustering, as well as with previously reported estimates. The average Fst statistics over each sliding window built from 1000 SNPs were calculated. We used the two first PCs calculated based on genotypes and the density-based spatial method implemented in the sklearn Python package (DBSCAN function) to determine potential subpopulations within the Polish population.

### 4.8. Pathogenic Allele Burden

The list of 2512 autosomal recessive (AR) disease-associated genes was obtained from the OMIM database (see Web Resources). Variants located within the coding regions as well as +/−5 kbp regions before and after the start and end of the coding sequences of these genes were extracted from the POL and gnomAD v3.0 cohorts and annotated using VEP (release 97), including ClinVar annotation (release date 17-05-2021). Of those, we selected nonsynonymous and regulatory regions (VEP IMPACT = ‘HIGH’, ‘MODERATE’, or ‘MODIFIER’) and pathogenic or likely pathogenic (according to ClinVar) variants without conflicting interpretation. For each of the 2463 genes with pathogenic variants in either POL or gnomAD NFE, we calculated the counts of variant alleles in the cohorts. Fisher’s exact test with FDR correction for multiple testing was used to test whether the cumulative frequencies in particular genes differed between the cohorts. Genes covered in POL by less than 20× in more than 10% of the canonical transcript length were excluded from the analysis (*n* = 41; Appendix A).

In addition to the small variant frequencies, we analysed possible loss-of-function (LoF) structural variants in the AR genes, using LoF prediction criteria following Collins et al. 2020 [42]. In brief, deletion was regarded as LoF when there was any overlap with any exon. In case of duplications, the additional criteria were that both breakpoints must be localised within a gene. For inversions, the same rules as for duplications were applied with the addition of SVs that had exactly one breakpoint within the gene. Similarly as for small variants, we calculated the counts of LoF alleles in each gene in the POL and gnomAD-SV v2.1 EUR cohorts. SVs with a frequency above 1% in each of the cohorts were excluded as likely benign.

Gene panels were downloaded from the PanelApp website (see Web Resources), selecting high-confidence gene–disease associations (annotated as “expert review green”) with the mode of inheritance defined as “BIALLELIC”. For each of the 325 panels, we summed the counts of pathogenic alleles for all AR genes in the panel. Similarly to the gene-level analysis, we tested the difference between POL and gnomAD NFE populations using a Fisher’s exact test with correction for the number of recessive genes in each studied gene panel and FDR correction for multiple testing (q-value).

## 5. Conclusions

We present the results of the whole-genome sequencing and comprehensive analyses of a wide spectrum of genomic variation in 1222 individuals of Polish ancestry. We detected and genotyped small and structural variants (SV), runs of homozygosity, mitochondrial haplogroups, and *de novo* variants. PCA and ADMIXTURE analyses indicated that the Polish population is homogenous and shares ancestry with other European populations. We also identified disease-causing alleles enriched and depleted in the Polish population. Allele frequencies of the unrelated individuals (1076) were released as a publicly available resource, The Thousand Polish Genomes database. To our knowledge, this is the largest collection of allele frequencies for a single Slavic population made available for research to date.

**Web Resources:** The Thousand Polish Genomes database, https://1000polishgenomes.com (accessed on 20 March 2021).

**FastQC v0.11.7:**https://www.bioinformatics.babraham.ac.uk/projects/fastqc/ (accessed on 20 March 2021.

**Smoove v.0.2.6:**https://github.com/brentp/smoove (accessed on 20 March 2021).

**Omim Database:**https://www.omim.org/ (accessed on 26 April 2021).

**PanelApp Website:**https://panelapp.genomicsengland.co.uk/panels/ (accessed on 27 June 2021).

**Clinvar Database:**https://www.ncbi.nlm.nih.gov/clinvar/ (accessed on 26 April 2021).

**CFTR Mutation Database:**http://www.genet.sickkids.on.ca/StatisticsPage.html (accessed on 30 August 2021).

## Figures and Tables

**Figure 1 ijms-23-04532-f001:**
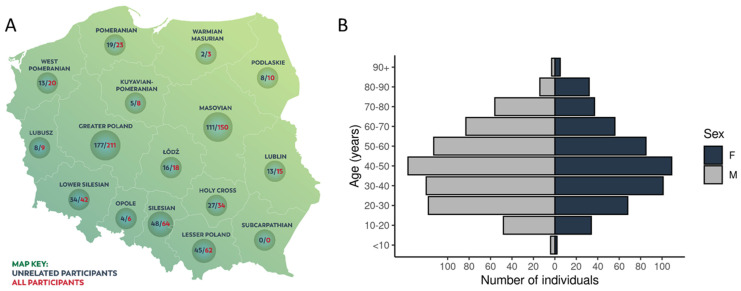
Polish cohort characteristics. (**A**) Sample collection locations for 675 individuals enrolled in “The Thousand Polish Genomes Project” (related and unrelated ones) in respect to Polish voivodeships. The figure presents a subset of samples with location data available. (**B**) Age distribution of 1076 unrelated study participants.

**Figure 2 ijms-23-04532-f002:**
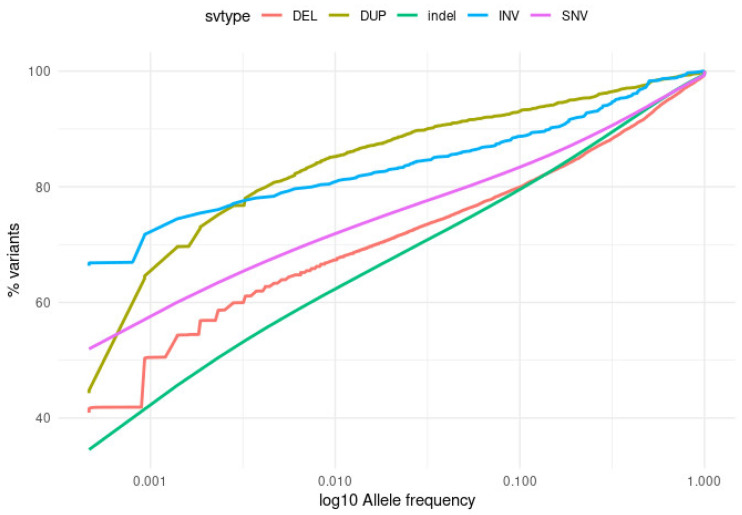
Distribution of allele frequencies across different variant types. DEL—large deletions; DUP—duplications; indel—short insertions and deletions; INV—inversions; SNV—substitutions. Variant impact categories follow VEP classification: HIGH—disruptive impact on the protein; MODERATE—non-disruptive variant that might change protein effectiveness; LOW—harmless or unlikely to change protein behaviour; MODIFIER—non-coding sequence variant.

**Figure 3 ijms-23-04532-f003:**
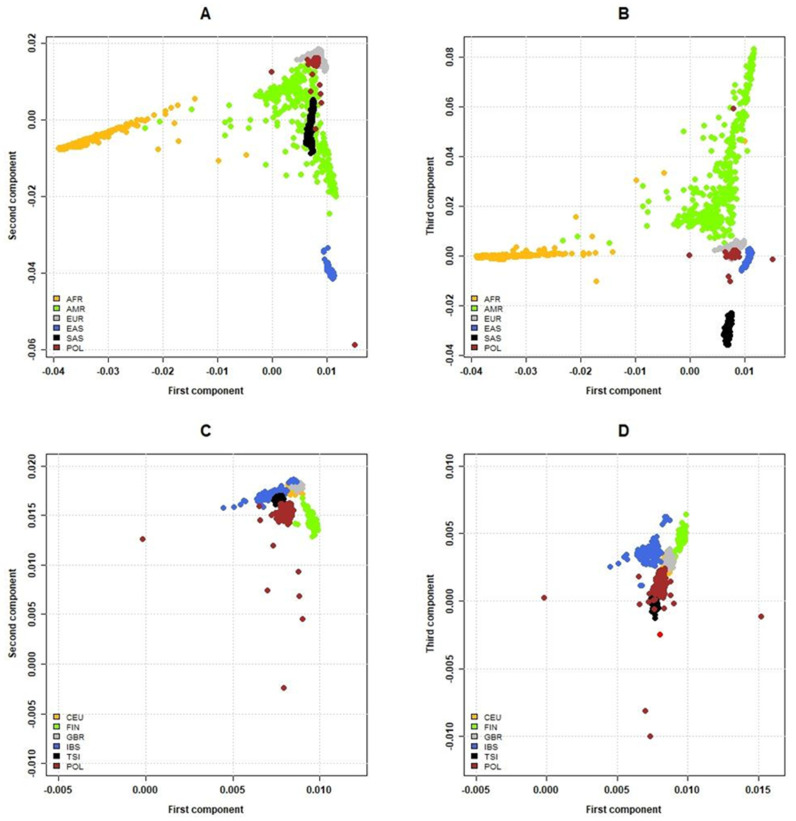
Clustering of samples among the global and European subpopulations based on the random forest and PCA predictions. Comparison of (**A**) the first and second principal components for continental populations, (**B**) the first and third principal components for continental populations, (**C**) the first and second principal components for the European subpopulations, and (**D**) the first and third principal components for the European subpopulations. Analysed populations: Utah residents (CEPH) with Northern and Western European ancestry (CEU), Finnish in Finland (FIN), British in England and Scotland (GBR), Iberian in Spain (IBS), and Toscani in Italy (TSI).

**Figure 4 ijms-23-04532-f004:**
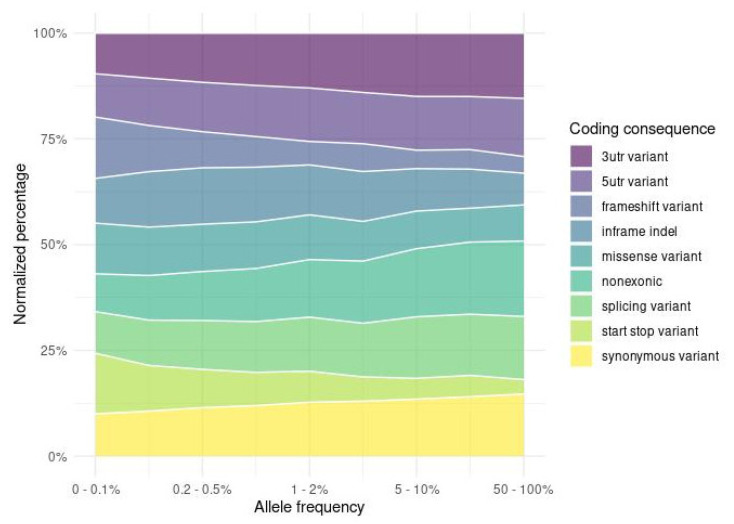
Comparison of exonic and non-exonic variant consequences across the allele frequency spectrum. As expected, variants with the largest impact on the encoded protein (e.g., start–stop, frameshift) were depleted among the common and enriched among the rare variation. On the opposite side, the relative abundance levels of non-exonic, UTR, and synonymous variants, which do not alter the amino acid sequence, increased with increasing variant population frequency.

**Figure 5 ijms-23-04532-f005:**
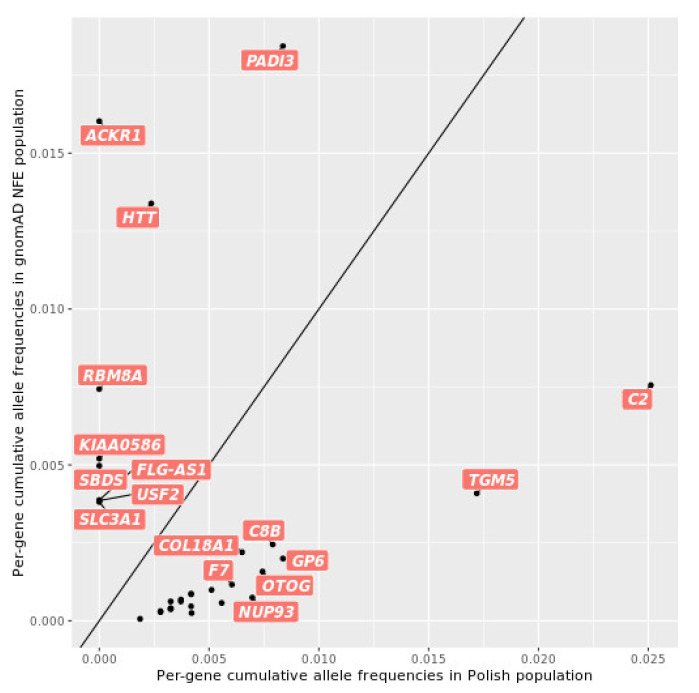
Cumulative allele frequencies of selected ClinVar variants in the 32 AR genes with significant (q-value < 0.05) differences in pathogenic variant burden among POL and gnomAD NFE. Note that for two genes with significant differences, the only variant observed in POL was removed from Clinvar after the release we used. These were *HTT* (rs754013273; RCV001335909.1) and *COL18A1* (rs528991245; RCV001329610.1).

**Table 1 ijms-23-04532-t001:** Summary of variant counts in three allele frequency tiers (>0.5%; 0.1–0.5%; <0.1%). Variant impact categories follow VEP classification: HIGH—disruptive impact on the protein; MODERATE—non-disruptive variant that might change protein effectiveness; LOW—harmless or unlikely to change protein behaviour; MODIFIER—noncoding sequence variant.

		IMPACT			
VARIANT_CLASS	AF	HIGH	MODERATE	LOW	MODIFIER
deletion	>0.5%	412	603	855	1,208,322
insertion		260	573	977	1,380,654
SNV		1109	35,717	41,402	10,877,171
deletion	0.1–0.5%	392	492	316	433,985
insertion		197	345	376	529,654
SNV		852	23,682	18,675	4,375,036
deletion	<0.1%	2849	1988	1003	1,295,678
insertion		1382	1144	826	1,037,730
SNV		5432	119,843	80,467	17,817,903
**TOTAL**		**12,885**	**184,387**	**144,897**	**38,956,133**

## Data Availability

The variant allele frequencies database is available to academic researchers through a simple application process and data usage agreement. We encourage qualified researchers and clinicians to visit the project website: 1000polishgenomes.com. Code Used to Analyse the Data for this Manuscript is available at https://github.com/MNMdiagnostics/NaszeGenomy (accessed on 27 June 2021).

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
