# Peer review of "The Thousand Polish Genomes—A Database of Polish Variant Allele Frequencies"

_ijms, 2022, doi:10.3390/ijms23094532_

Round 1

Reviewer 1 Report

Overall the manuscript is well written and of scientific importance both from a population genetics point of view and from a biomedical point of view. 

I consider that some graphics need to be reformulated as are quite confusing and not self-explained as they should be.

Further comments are in the pdf file attached.  

Reviewer 2 Report

The authors sequenced a large cohort of Polish population with great sequencing coverage. They genotyped different structural variants, mitochondrial haplotypes and performed Mendelian analyses. They also put the Polish results in context by comparing with other global populations.

The methods are well described and are standard. The limitations of the cohort are presented and discussed (not representative of the entire Polish population).

The work in this study represents a great resource, including genetic characterization of a Polish cohort, contributing with an open public database, with impact on the representation of diverse populations in human genomics and eventually with health-related research.

Minor concerns:

-It would be useful to clarify abbreviations of all populations used in the figure legends.

Reviewer 3 Report

The authors state several times “coverage” but they may mean “average read depth”. Please clarify
